# Luminescent Polymer Composites for Optical Fiber Sensors

**DOI:** 10.3390/polym15030505

**Published:** 2023-01-18

**Authors:** Rodolfo A. Carrillo-Betancourt, A. Darío López-Camero, Juan Hernández-Cordero

**Affiliations:** Instituto de Investigaciones en Materiales, Universidad Nacional Autónoma de México, A.P. 70-360, Ciudad de México 04510, Mexico

**Keywords:** PDMS fluorescent composites, optical fiber sensors, up-conversion, down-conversion

## Abstract

Optical fiber sensors incorporating luminescent materials are useful for detecting physical parameters and biochemical species. Fluorescent materials integrated on the tips of optical fibers, for example, provide a means to perform fluorescence thermometry while monitoring the intensity or the spectral variations of the fluorescence signal. Similarly, certain molecules can be tracked by monitoring their characteristic emission in the UV wavelength range. A key element for these sensing approaches is the luminescent composite, which may be obtained upon allocating luminescent nanomaterials in glass or polymer hosts. In this work, we explore the fluorescence features of two composites incorporating lanthanide-doped fluorescent powders using polydimethylsiloxane (PDMS) as a host. The composites are obtained by a simple mixing procedure and can be subsequently deposited onto the end faces of optical fibers via dip coating or molding. Whereas one of the composites has shown to be useful for the fabrication of fiber optic temperature sensors, the other shows promising result for detection of UV radiation. The performance of both composites is first evaluated for the fabrication of membranes by examining features such as fluorescent stability. We further explore the influence of parameters such as particle concentration and density on the fluorescence features of the polymer blends. Finally, we demonstrate the incorporation of these PDMS fluorescent composites onto optical fibers and evaluate their sensing capabilities.

## 1. Introduction

Detection and quantification of biological and chemical species are required in several areas, such as environmental science, clinical diagnostics, biology and chemistry. Among the available options for performing these tasks, optical methods offer several advantages over other sensing approaches, the most important of which are their high sensitivity and selectivity, both of which are highly desired features in biochemical sensing systems [1,2]. Although several techniques for optical sensing have been demonstrated, fluorescence-based measurements are recognized as among the most powerful tools for biochemical sensing [1,2,3]. Aside from providing remote monitoring capabilities, fluorescence signals can be analyzed in different ways, allowing for the realization of a wide variety of sensing schemes. Spectral features, lifetime and intensity are perhaps the most common examples of the different aspects that can be analyzed from a fluorescent signal. The advent of compact and rugged optical instruments such as solid-state spectrometers, as well as the availability of optical fibers, light sources and detectors, has enabled the development of compact and portable fluorescent sensing platforms, exploiting the versatility offered by fluorescence sensing.

Although some analytes provide suitable fluorescence emission for sensing, external fluorescent indicators (i.e., exogenous fluorophores) may be added to the samples if the target molecule lacks endogenous fluorophores [3]. Nanoparticles of different kinds, gold surfaces, carbon and quantum dots have been combined with chemical receptors, yielding fluorescent materials and enabling fluorescence detection [4]. Fluorescent polymers obtained either by incorporation of fluorophores in the polymer chain or by aggregation (so-called aggregation-induced emission polymers) have attracted a considerable amount of attention, owing to their viscoelastic and mechanical properties that render them useful for device fabrication [3,4]. Alternatively, fluorescent polymer composites may be realized upon incorporation of fluorescent dyes or nanoparticles into nominally transparent polymer matrices, either by adsorption, covalent binding or encapsulation [5]. These fluorescent composites have been widely used to obtain thin films and microspheres for sensing applications [3,4,5,6]. Aside from being simple to obtain, these materials can be conveniently incorporated onto optical fibers and waveguides for sensor fabrication. Both planar waveguides and optical fibers are key components in compact fluorescence sensing platforms; fibers, in particular, allow for spectroscopic measurements to be performed at remote locations and over a wide spectral range. When coated with adequate fluorescent polymers, they also allow for the detection of molecules that are otherwise imperceptible with optical spectroscopy [7,8]. Moreover, the availability of fiber bundles and microstructured fibers (e.g., photonic crystal fibers) has offered novel means for the development of devices for fluorescence imaging, as well as sensor arrays [7,8,9,10,11]. The combination of novel fiber optic structures and fluorescent polymer composites has thus increased the potential of fiber optic sensors in biochemical and biomedical applications.

Among the different polymer hosts reported for fluorescent dyes and nanoparticles, polydimethylsiloxane (PDMS) offers attractive features for fluorescence sensing. This silicone elastomer has been widely used in several areas, owing to its remarkable thermal, rheological, mechanical and biological properties [12,13]. Because the fabrication process of microfluidic chips is commonly based on soft lithography in PDMS, it is also sought as the building block for microfluidic platforms [13], and because it is optically transparent (240–1100 nm), it has also been used for the fabrication of planar waveguides and optical fibers [14,15]. Furthermore, PDMS can host fluorophores, thereby enabling the fabrication of microfluidic arrangements with fluorescence sensing capabilities [2]. In this paper, we explore the use of PDMS fluorescent composites for the fabrication of optical fiber sensors. We focus on two composites incorporating lanthanide-doped fluorescent powders using PDMS as a host. Whereas one of the composites has been shown to be useful for fabricating fiber optic temperature sensors, the other shows promising results for detection of UV emission. Temperature sensing is relevant for a myriad of applications, and UV-fluorescence detection is relevant for biosensing and biochemical applications. We first describe the procedure for obtaining the composites based on simple mixing of fluorescent powders with PDMS. The performance of both composites is first evaluated for fabrication of membranes and tested for features such as fluorescent stability under different conditions. Finally, we demonstrate the incorporation of these PDMS fluorescent composites onto optical fibers and evaluate their sensing capabilities. In contrast to previous reports using a custom-made dual-tip optical fiber device for temperature sensing [16,17], we deposit the temperature-sensitive composite on the tip of a fiber with a double-cladding structure, yielding a simpler sensor tip. In addition, the obtention of a UV-fluorescence-sensitive composite and its subsequent incorporation on the tips of conventional optical fibers provide a simple means to realize compact fluorescence sensors for this wavelength range.

## 2. Materials and Methods

Polymer composites with lanthanide-doped fluorescent powders were obtained following a mixing procedure first reported for the fabrication of photothermal membranes [18]. The composites are based on the use of Dow Corning Sylgard 184 PDMS as a host for the fluorescent materials. For our experiments, two polymer composites were prepared: one using an up-conversion (UC) phosphor (NaY_0.77_Yb_0.20_Er_0.03_F_4_, Sigma Aldrich 756555-25G, Saint Louis, MO, USA) and a second one using europium-activated oxysulphide down-conversion phosphor (Gd_2_O_2_S:Eu, Phosphors Technology, Stevenage, UK, UKL63/F-U1). UC ytterbium- and erbium-doped sodium yttrium fluoride powder has a particle size of 1–5 μm and is optimized to absorb in the wavelength range of 940–980 nm; the resulting PDMS-YbEr composite has been previously demonstrated in fiber optic temperature sensors [16]. Oxysulphide powder (GOS:Eu) has a mean particle size of 2–3 μm, exhibits a wide excitation band located in the UV region with a maximum absorption peak around 345 nm and has been shown to be useful for coatings in sensing and imaging applications [19,20]. As described elsewhere [21] and as shown in the following sections, the resulting PDMS-GOS:Eu composite provides attractive features such as a UV-fluorescence sensing material for fiber optic sensors.

The composites were fabricated following the same process, adding chloroform into the mix in order to reduce the viscosity of the silicone oil of the PDMS. The methodology is illustrated in Figure 1 and involves the following steps:(a)Weigh the phosphors and place them in a beaker;(b)Pour the base materials of the PDMS elastomer (silicone oil) and the chloroform (CHCl_3_) into the beaker (1 g/1.5 mL of PDMS/CHCl_3_ ratio);(c)Place the beaker in a combined hot-plate/magnetic-stirrer device for 60 min at 80 rpm and 90 °C to obtain a homogenous mix and to evaporate the chloroform;(d)Remove the beaker from the hot plate/magnetic stirrer and leave it to rest until reaching room temperature (approximately 10 min). Then, add the PDMS curing agent (10:1 ratio) and mix manually for 1 min;(e)Place the mixture in a vacuum chamber for degassing for 60 min;(f)Pour the PDMS-phosphor mixture in the desired substrate (either molds or glass slides (see below)) to obtain membranes;(g)Finally, place the substrate with the PDMS-phosphor composite in an oven at 80 °C for 4 h for solidification.

To evaluate the fluorescence features of the polymer composites, two different fabrication methods were explored. The PDMS-YbEr membranes were obtained by pouring the mix into molds fabricated from glass substrates and using height spacers approximately 150 μm thick. After pouring, the excess material was removed using the doctor blade technique. With this procedure, we obtained membranes with three different phosphor concentrations (1.5, 2.0 and 2.5% wt./wt.) and two different thicknesses (150 and 300 μm). In contrast, the PDMS-GOS:Eu mix was poured onto glass substrates, and membranes were obtained by means of a conventional spin coater. Upon adjusting the settings of the spin coating process, we were able to obtain membranes with thicknesses up to 350 μm and phosphor concentrations of 0.1, 0.5, 2.5, 12.5 and 62.5% wt./wt. The fabrication method used to obtain the membranes (i.e., molding or spin coating) was chosen in view of the approach followed to incorporate the composites onto the optical fiber devices, whereas for the PDMS-YbEr, we used a molding process, and the PDMS-GOS:Eu was incorporated through dip coating (see the following sections). Note that both methods can be used interchangeably for any of the polymer blends, and this choice should not affect their fluorescence features nor their sensing performance.

## 3. Polymer Composite Membrane Characterization

The fluorescence features of PDMS-phosphor composites with different concentrations and membrane thickness were evaluated. Aside from the spectral features, parameters such as thermal stability and fluorescence intensity were analyzed prior to their incorporation onto optical fibers.

### 3.1. PDMS-YbEr Composite

Optical microscopy images of the PDMS-YbEr membranes are shown in Figure 2. Although clusters of ytterbium- and erbium-doped sodium yttrium fluoride particles are evident from the images, the concentrations and thicknesses used to obtain the membranes do not seem to be increased. The fluorescence emission from PDMS-YbEr composites varies with temperature, and owing to its ease of processing, it is a suitable material for temperature sensing [16,17]. Using the setup shown in Figure 3, we evaluated the changes in fluorescence from the membranes as a function of temperature. This characterization is based on analysis of images from the PDMS-YbEr membranes acquired at different temperatures under fixed irradiation conditions. As depicted in the figure, the setup includes a fiber-coupled laser diode (LD, 975 nm, 330 mW maximum output power), a temperature controller to adjust the temperature of a ceramic heater through a thermoelectric cooler (TEC) and a CCD camera with an infrared (IR) filter to obtain fluorescence images from specific zones of the samples (i.e., the membranes).

Typical images obtained when irradiating one of the membranes (300 μm thickness, 2.5% concentration) with 140 mW of optical power and for different temperatures are included in Figure 4a–c. Clearly, the fluorescence from the membranes decreases with temperature. A direct relation between the temperature and the intensity of the fluorescence emitted by the PDMS-YbEr membranes can be obtained by means of image analysis. For this task, we used a technique originally developed for laser-induced fluorescence thermometry (LIFT) to acquire temperature maps with micron-scale spatial resolution [22]. Briefly, images were acquired at different temperatures using the setup shown in Figure 3; these were subsequently analyzed through direct comparison between a reference image and images obtained at prescribed temperatures. In our case, the reference image was obtained at room temperature, and its average intensity was used to normalize the average intensities acquired at specific temperatures set with the TEC (see [22] for details). This procedure allows for the obtention of a calibration curve for each membrane, yielding a plot of normalized intensity as a function of temperature. A typical curve showing the average changes in intensity as a function of temperature is included in Figure 4d. The plot was obtained from three sets of experiments (labelled Exp 1 to Exp 3 in the plot) and clearly shows a linear decrease in intensity as a function of temperature. A linear fit of the experimental points provides information regarding the sensitivity and the linearity of the registered variations in fluorescence intensity; the results obtained for all the fabricated membranes of PDMS-YbEr composites are summarized in Table 1. Whereas the linearity is very similar for all the tested samples, an increase in concentration seemingly yields a decrease in sensitivity. At a fixed concentration, the sensitivities decrease slightly for thicker membranes. Because the membranes were irradiated using the same optical powers, this behavior could be attributed to an increase in optical absorption associated with a larger optical path for thicker membranes. In addition, the temperature gradient might also change slightly when increasing the membrane thickness. Nonetheless, the performance of the PDMS-YbEr composites is practically sustained for the experimental conditions used for this characterization.

We further explored the thermal stability of the membranes when exposed to a fixed temperature for time periods of up to one hour. The membranes were irradiated with a fixed optical power (140 mW), and the temperature was set at a fixed value by means of the TEC. As shown in Figure 5, the average intensities registered at the prescribed temperatures remain stable throughout the full duration of the tests (similar trends were observed for the other fabricated membranes). Furthermore, in agreement with the results shown in Figure 4, the registered fluorescence intensities are lower at higher temperatures. We also performed the same test after storing the membranes for several months with similar results, suggesting that the fluorescence features of the PDMS-YbEr composites are maintained over long periods of time. Thus, the membranes do not show evidence of thermal bleaching, nor is photobleaching evident, at least under our experimental conditions. These features are of interest for fluorescent thermometry, which typically relies on fluorescent dyes such as Rhodamine B (RhB) as the temperature-sensitive material (see [22] and references therein). Photobleaching and thermal bleaching hinder the performance of RhB in fluorescent thermometry applications, requiring recalibration procedures over time. UC phosphors thus provide improved thermal stability compared to fluorescent dyes. For the temperature ranges used in our experiments (up to 100 °C, limited by the TEC) the PDMS-YbEr membranes perform well, although thermal degradation occurs at higher temperatures. The temperature limit is set by the polymer host, i.e., the PDMS, which is known to degrade at approximately 300 °C [13]. Moreover, UC phosphors have been incorporated in glass matrices and have been shown to withstand temperatures within the same range [23]. In general, glass matrices are preferred over their polymer counterparts for temperature sensors, but as discussed earlier, polymers offer ease of processing and molding capabilities, making them attractive for the rapid fabrication of devices.

It is clear that PDMS-YbEr membranes can provide a means to obtain fluorescence temperature measurements based on intensity. However, the most attractive features of these composites for temperature sensing are related to their spectral characteristics. They have been shown to be suitable for incorporation in optical fiber-based devices in which temperature measurements are obtained through spectral analysis [16,17], details of which will be discussed in the Applications and Discussion section.

### 3.2. PDMS-GOS:Eu Composites

The luminescence response of these composites is of great interest because they have shown great potential for monitoring UV emissions from biochemical species [20,21]. Because cluster formation and bulkiness can lead to a decrease in luminescence for these powders [19,24], we explored the performance of PDMS-GOS:Eu composites by fabricating membranes with different thicknesses. This was achieved through adjustments of the velocity (*v_c_*) and the spinning time (*t_vc_*) of the spin coater [21,25]; the thicknesses of the membranes were subsequently measured with a micrometer (IP65, 293-348-30, Mitutoyo). As shown in Figure 6, the thickness of the membranes decreases as *v_c_* and *t_vc_* increase. Furthermore, in general, increasing the concentration of the powder leads to a decrease in membrane thickness. However, an increase in thickness is noticeable when the concentration is changed from 2.5 to 12.5% wt./wt., followed by a subsequent decrease for a concentration of 62.5% wt./wt. Because the volume of the mix used to obtain the membrane should not affect the resulting thickness [26], this effect is attributed to the density of the blend. The densities for each mix obtained from calculations are included in Table 2, which show that a change in concentration from 2.5 to 12.5% wt./wt. yields an increase in density of 0.084 g/cm^3^—much greater than those obtained at other concentrations.

Adjustments to the fabrication parameters also affect the homogeneity of the phosphor particles within the membranes. As *v_c_* and *t_vc_* increased, membranes with lower particle concentrations per area were obtained. This is evidenced in the microscopy images included in Figure 7 obtained from membranes with a concentration of 2.5% wt./wt. with *t_vc_* ranging from 5 to 80 s, as well as for *v_c_* from 1000 to 4000 rpm. In contrast, an increase in particle concentration per area was observed upon increasing the phosphor concentration in the mix, as shown in Figure 8 for the thicker membranes (*v_c_* = 1000 rpm, *t_vc_* = 5 s), as well as for thinner membranes (*v_c_* = 4000 rpm, *t_vc_* = 80 s). Because the pressure exerted by the mix on the substrate is directly proportional to its density, it is reasonable to expect that thinner membranes will be obtained for denser polymer blends [25,26]. As shown in Table 2, increasing the phosphor concentration yields higher densities, as reflected by the trends observed in Figure 6 and Figure 8.

Fluorescence characterization was performed in a spectrofluorometer with a photomultiplier tube (Fluorolog-3, Horiba, FL3-22). Fluorescence signals from the membranes were obtained by placing them in a sample holder oriented at 45° of the optical axes of the instrument (i.e., the beam trajectory). The light source and the detector were further oriented perpendicularly to each other in order to avoid straight incidence of the pump source onto the detector; therefore, only the fluorescent signal from the membranes was registered. The source was set at a wavelength of 340 nm in coincidence with the absorption peak of the phosphor. Figure 9a shows a typical spectrum obtained from the PDMS-GOS:Eu membranes; this spectrum, in particular, showed the largest peak intensity and was obtained from the thicker membranes (*v_c_* = 1000 rpm) fabricated with the maximum particle concentration (62.5% wt./wt.). In agreement with previous reports [19,20], the three main fluorescence peaks were located at 595, 616 and 625 nm. The effects of the spin-coater parameters (*v_c_* and *t_vc_*) on the peak intensities for each wavelength are shown in Figure 9a–c. In general, an increase in velocity and spinning time yields a decrease in peak intensities, which might be related to a decrease in membrane thickness. However, in all cases, for the membranes fabricated using *v_c_* = 1000 rpm, the intensity seems to increase when *t_vc_* is increased from 5 to 20 s.

Concentration effects on the peak fluorescence intensity can be observed in the plots included in Figure 10. Because all the peaks showed a similar trend, only the plots for the main peak (625 nm) are shown. Clearly, the fluorescence peak increases with concentration and, in general, decreases with higher velocities and longer spinning times. The maximum peak intensity was obtained for *v_c_* = 1000 rpm and *t_vc_* = 20 s, which suggests that these spin-coating parameters are optimum for membrane fabrication with PDMS-GOS:Eu composites. However, the trends observed in the rest of the plots show that, in general, a higher concentrations of phosphors yield higher emissions, and increasing velocity and spinning time decrease the emissions, owing to a reduction in membrane thickness.

Because composites are sought to be incorporated in optical fibers, we evaluated the feasibility of capturing the fluorescence emission from the PDMS-GOS:Eu membranes using a fiber bundle. Using the setup shown in Figure 11a, the membranes were irradiated with an LED (M340F3, Thorlabs, 340 nm peak wavelength), whereas the fluorescence features were analyzed by means of a solid-state spectrometer (CCS200, Thorlabs). The setup included a bifurcated fiber bundle (RP20, Thorlabs) intended for back-reflection spectroscopic measurements; this device incorporates seven fibers: one used to irradiate the samples and the remaining fibers to capture the fluorescence emission of the samples (see Figure 11a). Upon adjusting the current fed to the LED, the irradiation power was set at different levels, and the fluorescence spectra were captured through the common branch of the bifurcated fiber bundle. Figure 11b shows the normalized fluorescence spectra for two excitation powers (i.e., 13.5 and 14.4 μW). With our setup, we can detect the principal fluorescence peaks. Remarkably, in spite of the low optical power levels used to irradiate the membranes, the fluorescence emission reaches adequate values to be registered by the spectrometer.

## 4. Applications and Discussion

### 4.1. Optical Fiber Temperature Sensors with PDMS-YbEr Composites

As shown in the previous section, PDMS-YbEr membranes offer attractive features for fluorescence thermometry. The use of lanthanide (rare-earth) ions for fluorescence thermometry has been widely studied [27,28], and their incorporation in polymer matrices such as PDMS expands the possibilities for temperature sensor design. The fact that the fluorescence intensity of PDMS-YbEr membranes is temperature-dependent suggests that they may be useful in LIFT applications. For example, composites incorporating RhB as a temperature-sensitive dye could be replaced by the rare-earth composites, thereby avoiding fluorescence stability issues associated with photobleaching commonly encountered with fluorescent dyes [22]. However, the fluorescence spectrum of the PDMS-YbEr composites is perhaps the most attractive feature for temperature sensing. Techniques such as fluorescence intensity ratio (FIR) can be readily used for temperature measurements, as demonstrated with fiber optic devices [16,17]. Because some energy levels of the UC powder used in the composites are thermally coupled, changes in temperature differentially affect the corresponding spectral bands. Therefore, the temperature can be obtained according to the ratio of these emission bands, thereby minimizing spurious effects of the excitation laser and intensity fluctuations (see [16] and references therein). As described below, PDMS-YbEr composites can be readily incorporated in optical fibers with special structures to yield FIR-based temperature sensors.

A double-cladding fiber (DCF) structure can be used to realize a fluorescence temperature sensor. DCFs incorporate two concentric claddings with different refractive indices, effectively forming two waveguides in a single optical fiber. As depicted in the inset of Figure 12, a single-mode waveguide structure comprises the core (*Co*, refractive index *n_1_*) and the first cladding (*Cd_1_*, refractive index *n_2_*), whereas a multimode waveguide is formed by the first and second cladding (*Cd_2_*, refractive index *n_3_*). Each of these concentric cylinders has a different refractive index (*n_1_> n_2_ > n_3_)*; therefore, light is confined and guided within the different structures. Because the multimode structure has a larger cross-section, the fluorescence from the PDMS-YbEr composite can be effectively captured by this waveguide, and the central core can be used to excite the fluorescent material. The sensor is fabricated using a brass mold with a reservoir (10 mm length, 2 mm width and 2 mm depth) for the composite, with a groove to accommodate the DCF. Once the fiber is placed in the groove, the fluorescent composite is simply poured directly into the reservoir, and after degassing, everything is placed in an oven for curing (see Section 2 for curing settings). Once the PDMS-YbEr composite has solidified, the resulting sensor tip is demolded and can be readily calibrated. All the results shown here were obtained with sensors fabricated with PDMS-YbEr composite at a 2.5% concentration.

The spectral features of the fluorescence signal from the sensors were obtained using the setup depicted in Figure 12. The single-mode fiber (SMF) of a double-cladding coupler (DCC) is used to excite the PDMS-YbEr sensor tip, whereas the fluorescence is captured and sent through the multimode fiber (MMF) to a solid-state spectrometer (SS) for analysis. We used a fiber-coupled laser diode (LD) operating in the infrared range (*λ_IR_ = 975 nm*) to generate the green UC fluorescence signal (*λ_F_*) from the polymer composite, and spectra at different temperatures were obtained by placing the sensor tip on the ceramic heater and TEC (not shown). Typical spectra obtained from a sensor are included in Figure 13a, showing the characteristic emission bands of the UC phosphor [16,17]. Because the spectral bands labelled **H**_11/2_–**I**_15/2_ and **S**_3/2_–**I**_15/2_ are thermally coupled, they can be used to calculate the fluorescence intensity ratio (FIR) as a function of temperature (see [16] and references therein). Upon integrating both bands and measuring their ratios at different temperatures, we obtained the plot presented in Figure 13b, showing that the FIR varies linearly with temperature. The plot includes results from three different experiments; evidently, the calculated FIR for each temperature is repeatable, with an error within 0.2%. Furthermore, FIR varies linearly with temperature, and both the linearity and sensitivity (*S_T_* = 2.77 × 10^−3^ °C^−1^, *R^2^* = 0.999) are consistent with previous results obtained using multimode fiber sensors [16]. Similar results were obtained for different pump powers and for different probes fabricated following the same procedure, thereby validating the use of the FIR technique to eliminate noise and other spurious effects for fluorescence temperature measurements. Finally, we evaluated the ability of the sensors to track temperature changes when immersed in a liquid. This test was performed by immersing the sensing tip in water at room temperature (25 °C) and subsequently immersing the tip in a different container with water at a higher temperature. Results of this test involving switching the tip to containers with water at 50 °C and 70 °C are included in Figure 13c,d, respectively. For comparison, a commercial thermocouple was immersed along with the fiber sensor. The temperature readings from the two devices are very similar, providing evidence that the response time of the fluorescence fiber optic sensor is comparable to that of commercial thermocouples.

### 4.2. Devices with PDMS-UKL Composites

As shown in the previous section, PDMS-GOS:Eu composites offer attractive features for UV monitoring. The use of lanthanide ions for UV detection and monitoring through the down-conversion process has been widely studied [29,30,31,32]. Incorporation of these ions in polymer matrices such as PDMS provides a simple means to obtain a transducer that can absorb UV emitted in the visible portion of the optical spectrum. Evidently, other molecules with lanthanide ions could be used, but the emission of GOS:Eu allows for its fluorescence to be easily transmitted by conventional optical fibers. The sensitivity of these composite to UV further suggests that they may be useful in UV autofluorescence monitoring applications. For example, biological tests using external fluorescence labels, which require attached molecules with UV autofluorescence, could be eliminated and replaced by simple UV autofluorescence monitoring using these composites. This approach would not require external fluorophores to be disseminated in the samples and may also serve to eliminate the use of photomultipliers, which are required to register the generally weak autofluorescence signals [33].

PDMS-GOS:Eu composites can be easily incorporated onto optical fibers. For example, as shown for PDMS-YbEr composites, a mold can be used to attach the material onto an optical fiber tip. However, to explore other possibilities for sensor fabrication, we used a dip-coating technique to fabricate UV fiber optic sensors such as those included in Figure 14a,b. Using the dip-coating technique, devices can be fabricated with small drops of the polymer composites allocated on the tip of conventional fibers [34]. The size of the deposited polymer drop (i.e., composite thickness) is relevant because thicker coatings can lead to a decrease in the luminescence; Figure 14a shows a microscopy image of a multimode optical fiber coated with PDMS-GOS:Eu at 62.5% wt./wt., whereas Figure 14b shows the emission of the composite allocated on the fiber tip. The emission from the coating guided by the multimode fiber was registered with the setup shown in Figure 14c; the opposite end of the sensor tip was connected to a solid-state spectrometer (CCS200, Thorlabs), and the composite was irradiated with a UV light source operating at 340 nm. As shown in Figure 14d, the spectra showed features registered with the membranes fabricated with the same composite. We further varied the optical power of the light source from 55.84 to 661.28 μW, showing that the peak power of the main emission bands varies accordingly. With a power of 55.84 μW, only the principal fluorescence peak (located at 625 nm) is barely registered with the spectrometer, and with increased excitation power, the full spectrum can be readily observed. Again, the low excitation powers required to obtain a spectral readout from this fiber probe represent a promising result for the development of monitoring devices to track UV biological signals.

## 5. Conclusions

Polymer composites can be easily obtained by incorporating lanthanide-doped powders in PDMS, a material that has been widely used for various applications. In particular, NaY_0.77_Yb_0.20_Er_0.03_F_4_ and Gd_2_O_2_S:Eu powders were incorporated into the PDMS via mixing, and membranes of these composites (PDMS-YbEr and PDMS-GOS:Eu, respectively) were used to evaluate their fluorescence features. Both materials showed excellent thermal stability, and no evidence of photobleaching or thermal bleaching was apparent under our experimental conditions. Whereas the PDMS-YbEr composite was verified as an attractive material for temperature sensing applications by means of up-conversion emission, the PDMS-GOS:Eu composite shows potential for monitoring UV emission via down-conversion. Given the ease of processing of PDMS, both materials can be readily incorporated onto optical fibers to obtain fiber optic sensors, the response of which can be analyzed using compact instruments commonly used with fiber sensing arrangements. Herein, we demonstrated two methods for fabricating fluorescence optical fiber sensors: a simple molding procedure and a dip-coating process. In both cases, the polymer composites were attached to the tips of optical fibers, yielding a fiber probe for fluorescent temperature sensing (PDMS-YbEr) and a fiber probe for UV detection (PDMS-GOS:Eu). We expect that these polymer composite devices will be useful in biochemical and biomedical applications for temperature sensing and for monitoring of UV emissions from biochemical species.

## Figures and Tables

**Figure 1 polymers-15-00505-f001:**
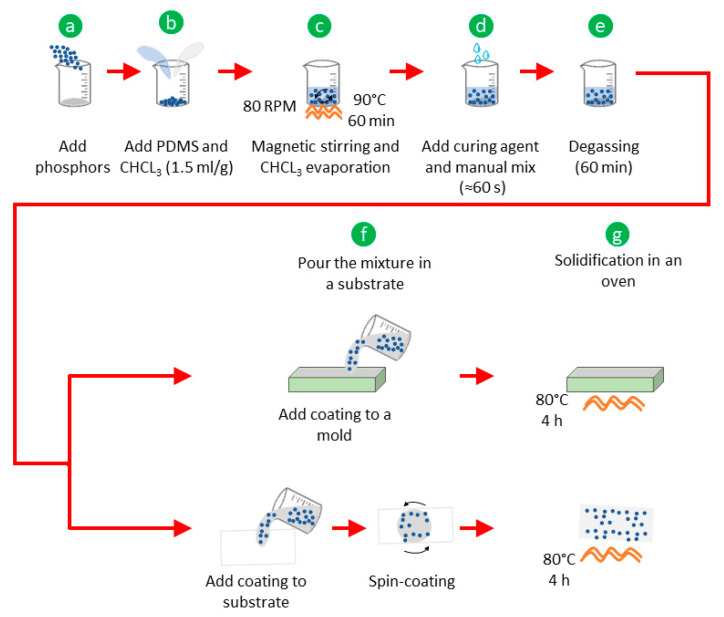
Mixing methodology to obtain the fluorescent polymer composites (PDMS-phosphor; see text for further details).

**Figure 2 polymers-15-00505-f002:**
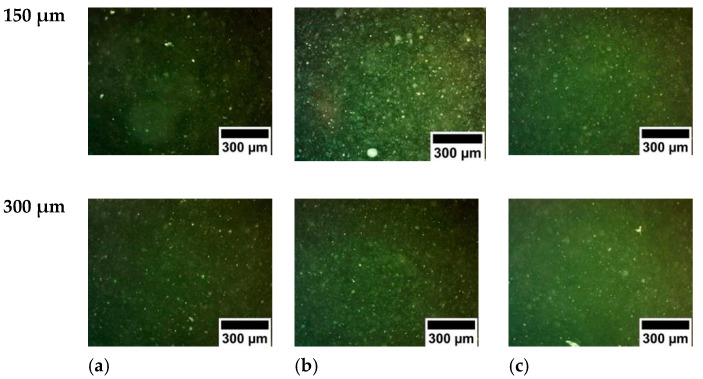
Optical microscopy images of PDMS-YbEr membranes with thicknesses of 150 μm (**upper row**) and 300 μm (**lower row**) and phosphor concentration of (**a**) 1.5%, (**b**) 2.0 and (**c**) 2.5%.

**Figure 3 polymers-15-00505-f003:**
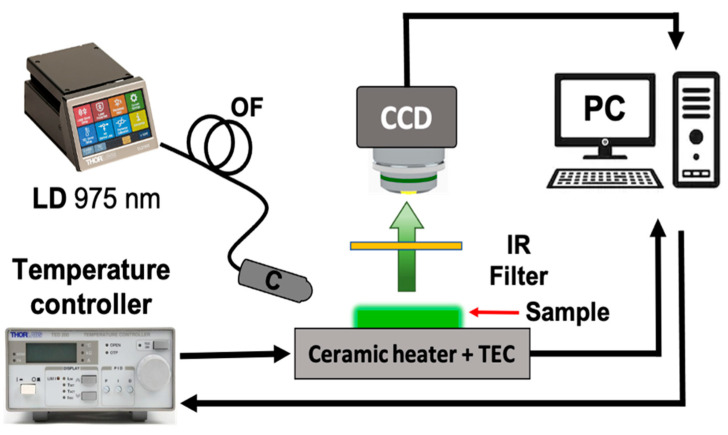
Schematic representation of the experimental setup used to analyze the fluorescence from the PDMS-YbEr membranes as a function of temperature: light from a laser diode (LD) coupled to an optical fiber (OF) is collimated (C) and used to irradiate the sample. The fluorescence from the membrane is captured by a CCD camera after filtering the residual IR radiation. Temperature adjustments are made by a TEC controller connected to a personal computer (PC), which is also for image acquisition.

**Figure 4 polymers-15-00505-f004:**
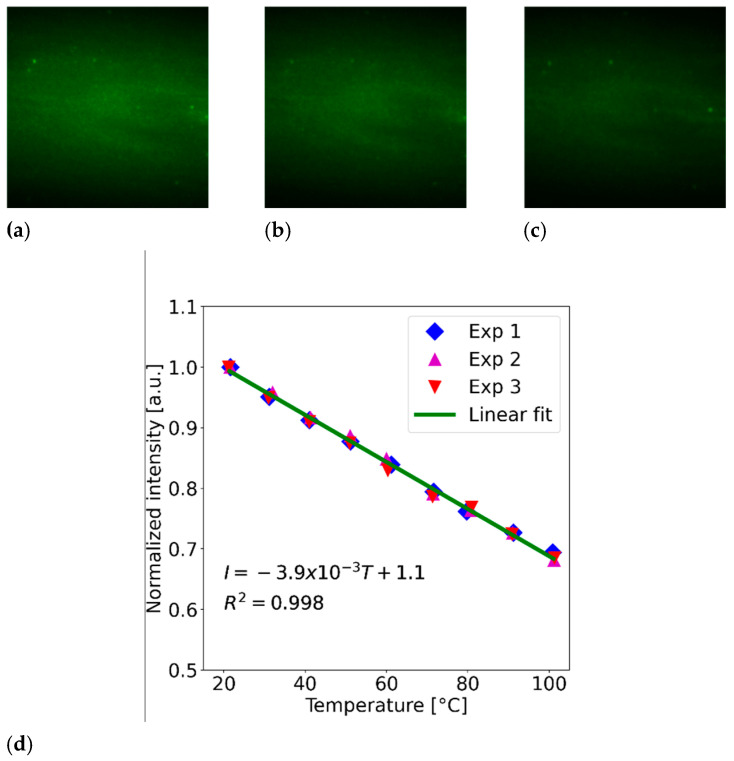
Typical fluorescence images obtained from a 300 μm thick membrane with 2.5% concentration (140 mW LD optical power) at temperatures of (**a**) 25 °C, (**b**) 50 °C and (**c**) 100 °C. (**d**) Changes in normalized intensity for the same membranes as a function of temperature; the dots indicate the experimental measurements obtained by image analysis, and the solid line represents the linear fit of the experimental points.

**Figure 5 polymers-15-00505-f005:**
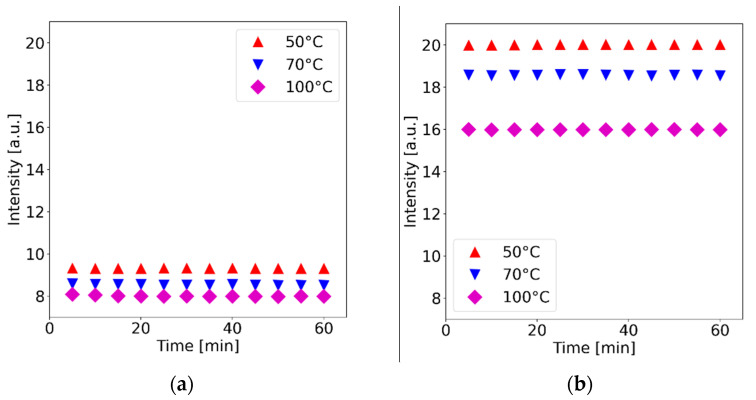
Fluorescence stability at different temperatures for PDMS-YbEr membranes of: (**a**) 150 μm thickness and 1.5% concentration and (**b**) 300 μm thickness and 2.5% concentration.

**Figure 6 polymers-15-00505-f006:**
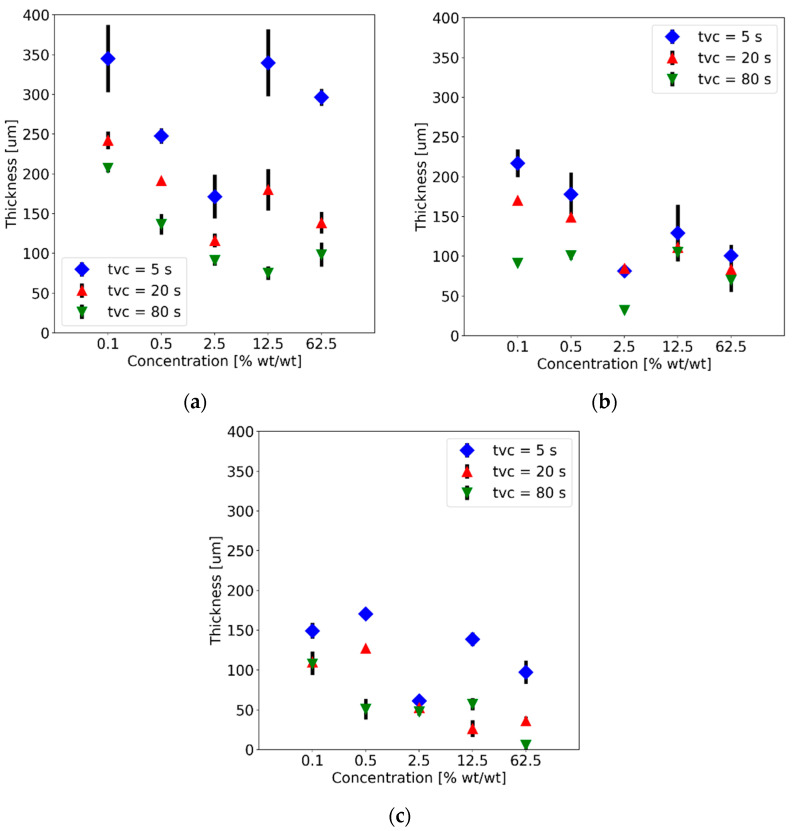
Membrane thicknesses obtained for membranes fabricated with different concentrations of PDMS-GOS:Eu powder. Results are presented for three different spin-coating velocities ((**a**) 1000 rpm, (**b**) 2000 rpm and (**c**) 4000 rpm) and for the indicated spinning times (*t_vc_*).

**Figure 7 polymers-15-00505-f007:**
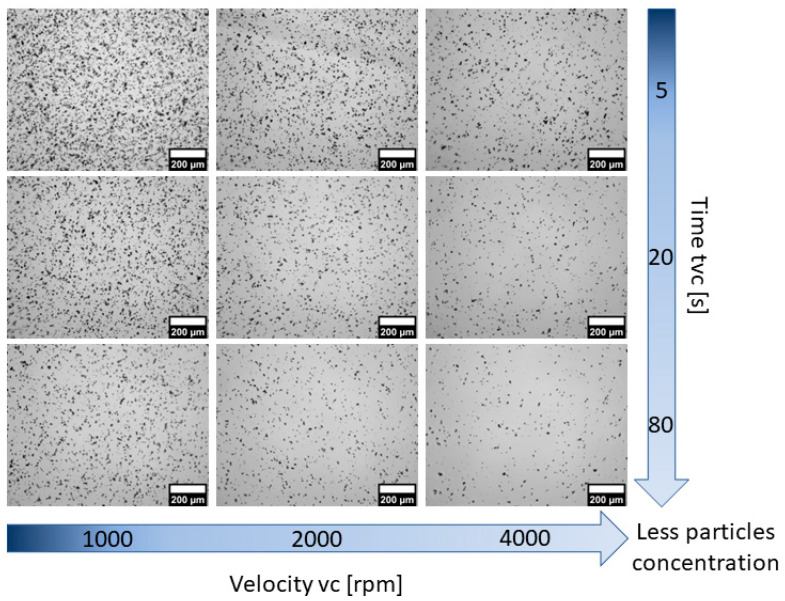
Optical microscopy images of the PDMS-GOS:Eu membranes (2.5% wt./wt. concentration) fabricated at different spin-coating velocities (*v_c_*) for different spinning times (*t_vc_*).

**Figure 8 polymers-15-00505-f008:**
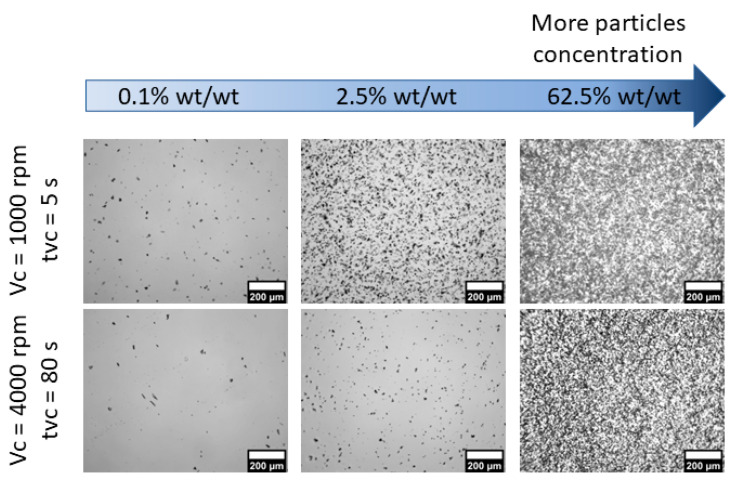
Optical microscopy images of membranes fabricated at 1000 rpm for 5 s (**upper row**) and at 4000 rpm for 80 s (**lower row**).

**Figure 9 polymers-15-00505-f009:**
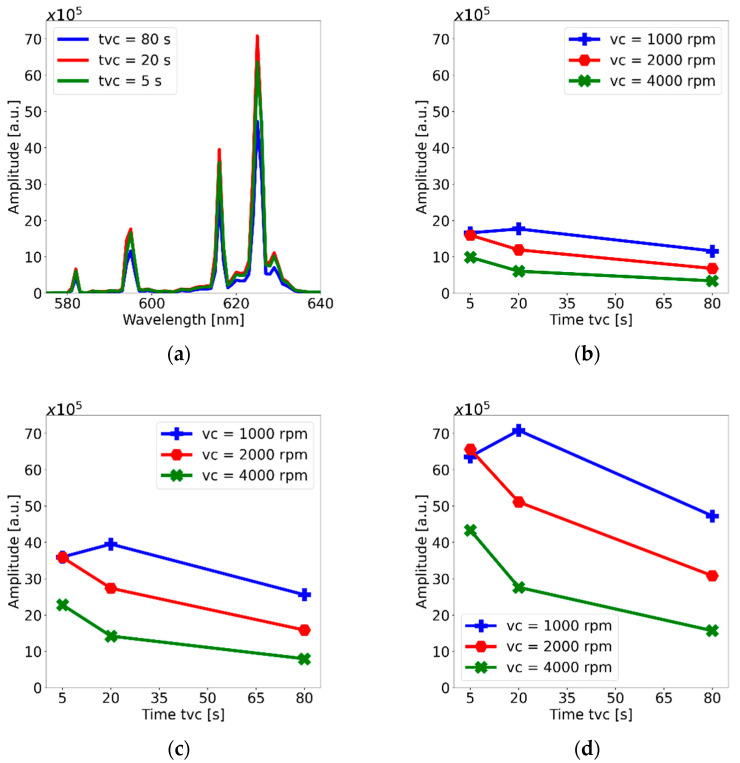
(**a**) Fluorescence spectra for the membranes with concentration of 62.5% wt./wt. fabricated at 1000 rpm. Changes in the three mean fluorescence peaks for membranes at concentration of 62.5% wt./wt.: (**b**) 595 nm, (**c**) 616 nm and (**d**) 625 nm.

**Figure 10 polymers-15-00505-f010:**
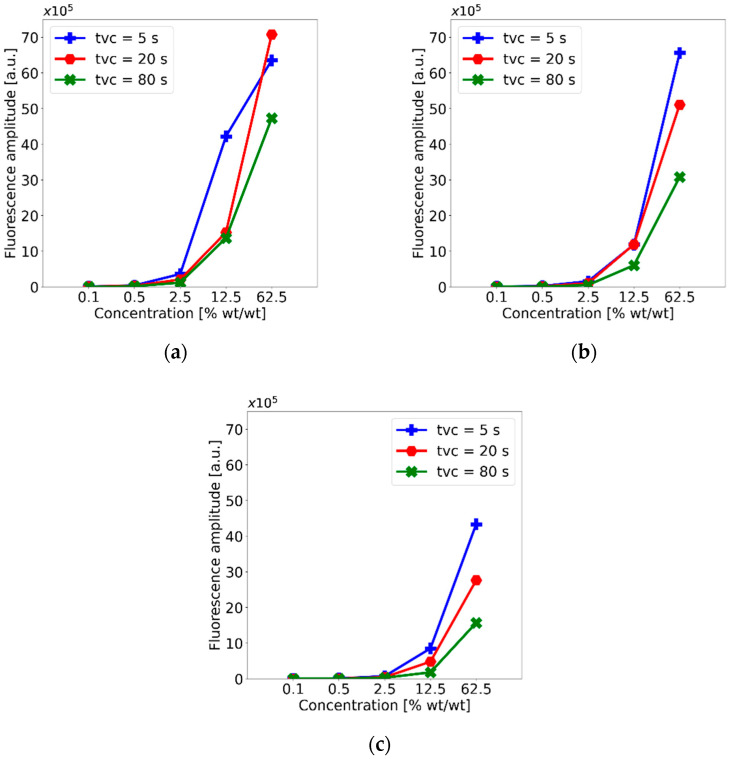
Variation in fluorescence peak intensity (625 nm) for different phosphor concentrations of the PDMS-GOS:Eu membranes fabricated using spin-coating velocities of (**a**) 1000 rpm, (**b**) 2000 rpm and (**c**) 4000 rpm.

**Figure 11 polymers-15-00505-f011:**
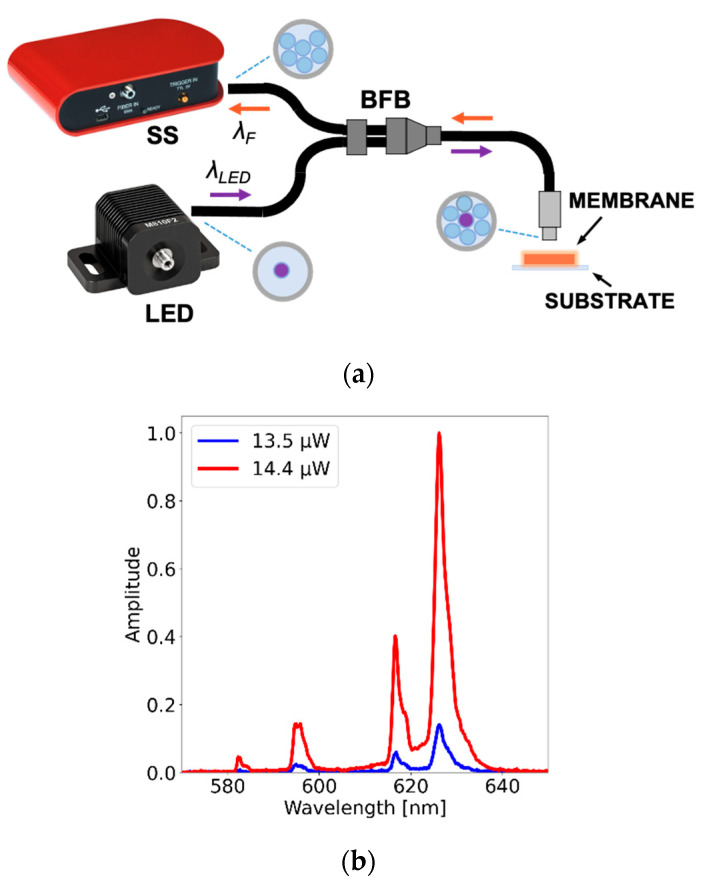
Typical spectral features of a PDMS-GOS:Eu membrane (62.5% wt./wt. concentration, *v_c_* = 1000 rpm, *t_vc_* = 20 s) as a function of the excitation power. (**a**) The setup used for spectral acquisition includes a bifurcated fiber bundle (BFB), an LED (340 nm) and a solid-state spectrometer (SS). (**b**) Fluorescence peaks captured with the bundle for two excitation powers (i.e., 13.5 and 14.4 μW) registered with the BFB (see text for further details).

**Figure 12 polymers-15-00505-f012:**
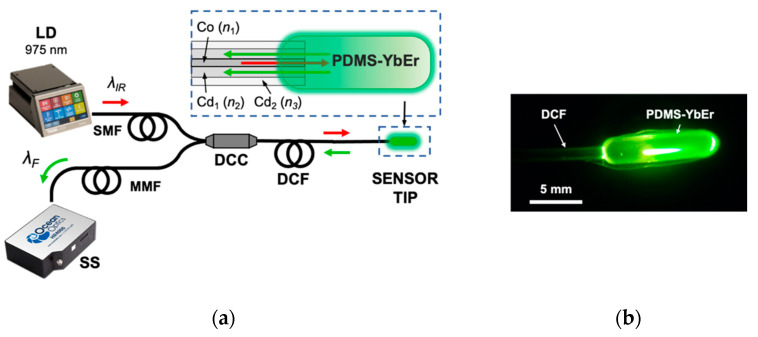
(**a**) Schematic representation of the experimental setup used to test an optical fiber temperature sensor. The sensing tip comprises PDMS-YbEr composite deposited on a double-cladding fiber (DCF). Using a double-cladding coupler (DCC), the fluorescent material is excited with the laser diode (LD, *λ_IR_*), and the fluorescence signal (*λ_F_*) is analyzed with a solid-state spectrometer (SS). The inset shows the structure of the DCF formed by a central core (Co) and two concentric claddings (Cd_1_ and Cd_2_). See text for further details. (**b**) Image of the sensor tip showing the green UC fluorescence from the PDMS-YbEr composite.

**Figure 13 polymers-15-00505-f013:**
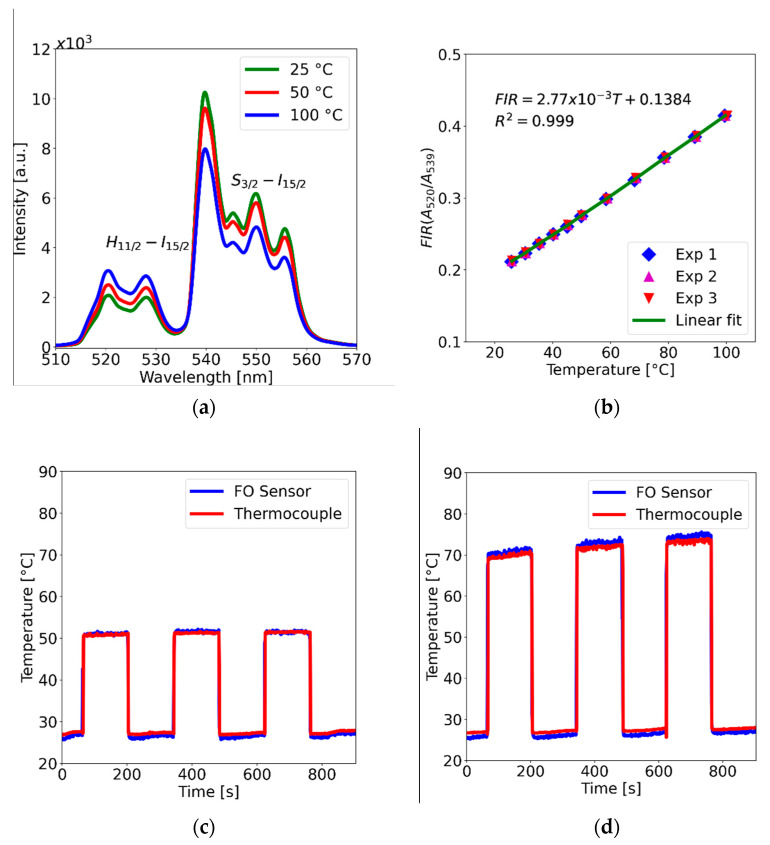
(**a**) Fluorescence spectra obtained from the PDMS-YbEr fiber tip (140 mW optical power); the bands labelled **H**_11/2_–**I**_15/2_ and **S**_3/2_–**I**_15/2_ are thermally coupled, and their ratio is used to obtain the temperature. (**b**) Fluorescence intensity ratio (FIR) as a function of temperature. (**c**,**d**) Readout of the fluorescence temperature sensor compared with a thermocouple sensor.

**Figure 14 polymers-15-00505-f014:**
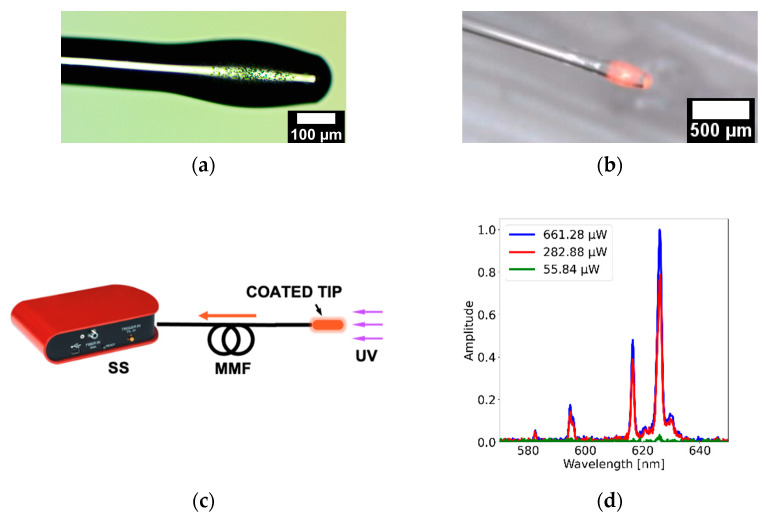
Multimode optical fiber coated with GOS:Eu: (**a**) microscopy image of the coated fiber; (**b**) coated tip showing the emission of the polymer coating; (**c**) setup used to obtain the fluorescence emission from the sensor tip as a function of optical power; (**d**) emission spectra obtained with irradiation from a UV light source (340 nm) using different powers (55.84 to 661.28 μW).

**Table 1 polymers-15-00505-t001:** Temperature sensitivities (*S_T_,* °C^−1^) and linearity (*R^2^*) of the normalized fluorescence intensities obtained for the PDMS-YbEr composite membranes.

Sample	d = 150 μm	d = 300 μm
C = 1.5%	*S_T_* = −5.2 × 10^−3^, *R^2^* = 0.996	*S_T_* = −5.0 × 10^−3^, *R^2^* = 0.995
C = 2.0%	*S_T_* = −4.8 × 10^−3^, *R^2^* = 0.995	*S_T_* = −4.4 × 10^−3^, *R^2^* = 0.991
C = 2.5%	*S_T_* = −4.8 × 10^−3^, *R^2^* = 0.991	*S_T_* = −3.9 × 10^−3^, *R^2^* = 0.998

**Table 2 polymers-15-00505-t002:** Concentrations and densities of the PDMS-GOS:Eu composites.

Concentration (% wt./wt.)	Density (g/cm^3^)
0.1	1.103
0.5	1.106
2.5	1.123
12.5	1.207
62.5	1.595

## Data Availability

The data presented in this study are available upon request from the corresponding author.

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
