# Peer review of "Luminescent Polymer Composites for Optical Fiber Sensors"

_polymers, 2023, doi:10.3390/polym15030505_

Round 1

Reviewer 1 Report

The work is devoted to the creation of luminescent polymer composite materials suitable for the application if optical fiber sensors. The composites are obtained by a simple mixing procedure and can be subsequently deposited onto the end faces of optical fibers via dip coating or molding. The authors focus on two composites incorporating lanthanide-doped fluorescent powders using PDMS as a host. While one of the composites has shown to be useful for fabricating fiber optic temperature sensors, the other one shows promising results for detection of UV emission. In general, the material is presented in an accessible language with a large amount of experimental data obtained by various methods. The work can be recommended for publication, but after the elimination of a number of issues. Comments are below:

  1. The authors throughout the manuscript actively refer to their previous works (in particular refs 16 and 20). These works describe a composite based on the same rare earth complex used in an optical temperature sensor. The authors should clearly indicate in the introduction the scientific novelty of this manuscript. What is new compared to previously published works?
  2. In the introduction, the authors explain in detail the reasons why the optical sensing is preferable in biochemical sensing systems. They also describe in detail the advantages of fiber optic measurement methods. Among the options for the introduction of the exogenous fluorophore, they indicate two types: nanoparticles of different kinds and fluorescent polymer composites. The reviewer fully agrees with the authors in this part, however, the sentence "Although in general these fluorescent composites are sought as unstable materials for long time monitoring" seems unreasonable. Composites are used for long-term measurements; moreover, they often make it possible to enhance the properties of the sensing material, for example, improving the signal intensity due to the introduction of metal nanoparticles, controlling the adhesion of the material, etc. Recent reviews can be cited as an example [10.3390/polym14204448, 10.1007/s10311-022-01403-2].

3.     The authors write: "While the linearity is very similar for all the tested samples, an increase in concentration seemingly yields a decrease in sensitivity. For a fixed concentration, the sensitivities decrease slightly for thicker membranes". It would be helpful to the reader if the authors suggested reasons for this behavior.

Minor issues:

  1. Add a subscript for the 4 in the formula "NaY0.77Yb0.20Er0.03F4"
  2. Perhaps it is worth bringing colored markers in Figure 4d, and/or indicating the average values and resulting errors when constructing the calibration dependence. The same for Figure 13b.
  3. Please color Figures 11b, 12b, and 14 a, b, d.

Reviewer 2 Report

Dear Editor,

            This manuscript investigated the fabrication of different types of Luminescent polymer composites based on lanthanide-doped polydimethylsiloxane (PDMS) for optical sensors. Have the following suggestions and comments to improve the state of the study. So, I recommend accept after some minor revisions. It is proposed to investigate the surface morphology of the composites using SEM or AFM analysis and can be added to the manuscript. Conclusion should be supported with more data results (numerical).
